# Intranasal cocaine self-administration in male mice

Kirsty R. Erickson [1], Yizhen Quan[1], Zahra Z. Farahbakhsh [1],
Hannah E. Branthwaite [1], Keaton Song [1], Justin D. Kim[1], Janice J. Lee[1],
Katherine N. Gibson-Corley [2], Eyal Y. Kimchi [3] & Cody A. Siciliano [1] ✉

Intravenous drug self-administration has been widely used in behavioral neuroscience to model addiction and heightened motivational states; however, the technique is notoriously difficult and its utility has been in steady decline. Here, circumventing prior issues, we established a procedure for intranasal drug self-administration in head-restrained male mice. This procedure does not require surgical expertise or indwelling implants, drives robust schedule- and dose-dependent responding, and results in high blood cocaine concentrations. In addition to improved ease of use and face validity for modeling cocaine use disorder, this work provides an example of volitional nasal drug insufflation by a non-human animal, a behavior canonically thought to be unique to humans across compounds.

Intravenous drug self-administration is the leading non-human animal model for the volitional use of narcotics and has been utilized extensively over the last century to investigate the circuit- and receptor-basis of motivation, decision-making, and addiction[1–3]. Though the unique motivational properties of drugs remain a powerful tool for addressing questions in modern neuroscience, intravenous drug self-administration is notoriously difficult. The requirement of significant surgical expertise for jugular vein catheterizations[4], high subject attrition (often greater than 50% in mice[5,6]), and difficulty integrating microfluidic hardware with modern neurotechnologies[7] has led to a steady decline in the utility of this approach. Surgical, catheter, and integration issues are a consequence of intravenous drug delivery, which is currently requisite to allow compounds to be self-administered in animals through a non-oral route. Furthermore, in humans, most narcotic users begin with intranasal insufflation[8–10]. While heavy opioid users often transition to intravenous use, intranasal administration is often the preferred route for cocaine users across recreational users and treatment seeking populations[11]. We reasoned that providing the means for intranasal drug consumption could circumvent these technical and conceptual issues.

Here, we have developed a procedure in mice that allows for operant self-administration of compounds through the intranasal route. We show that intranasal self-administration of cocaine engenders robust reinforcement, does not require surgical expertise, and can be readily integrated with contemporary techniques in behavioral neuroscience. To achieve this, an aluminum anchor rod was affixed to the skull with cranioplastic cement in a standard surgical procedure widely used for head-fixation in rodents[12] (see Methods). Upon recovery, in once-daily sessions, mice were head-restrained in the experimental apparatus, consisting of an elevated platform, minimal force lever, and anchoring posts (Fig. 1A). A microfluidic delivery system, consisting of a blunt needle mounted on a 3-axis miniature micromanipulator and fed by a high-resolution syringe pump, is used to deliver $0.5\,\mu L$ droplets of a cocaine solution (dissolved in saline) directly in front of the nostril of the head-restrained mouse. Delivery is contingent upon operant responding and, once delivered, can be easily insufflated by the subject (Supplementary Video 1).

## Results and discussion

### Intranasal cocaine functions as a reinforcer

First, we sought to test whether this configuration allowed cocaine to function as an intranasal reinforcer. To ascertain baseline response rates, mice were placed in the experimental apparatus during daily 30-minute sessions with the delivery device and lever

[1]Department of Pharmacology, Vanderbilt Brain Institute, Vanderbilt Center for Addiction Research, Vanderbilt University, Nashville, TN, USA. [2]Division of Comparative Medicine, Department of Pathology, Microbiology, and Immunology, Vanderbilt University Medical Center, Nashville, TN, USA. [3]Ken & Ruth Davee Department of Neurology, Feinberg School of Medicine, Northwestern University, Chicago, IL, USA. ✉e-mail: cody.siciliano@vanderbilt.edu

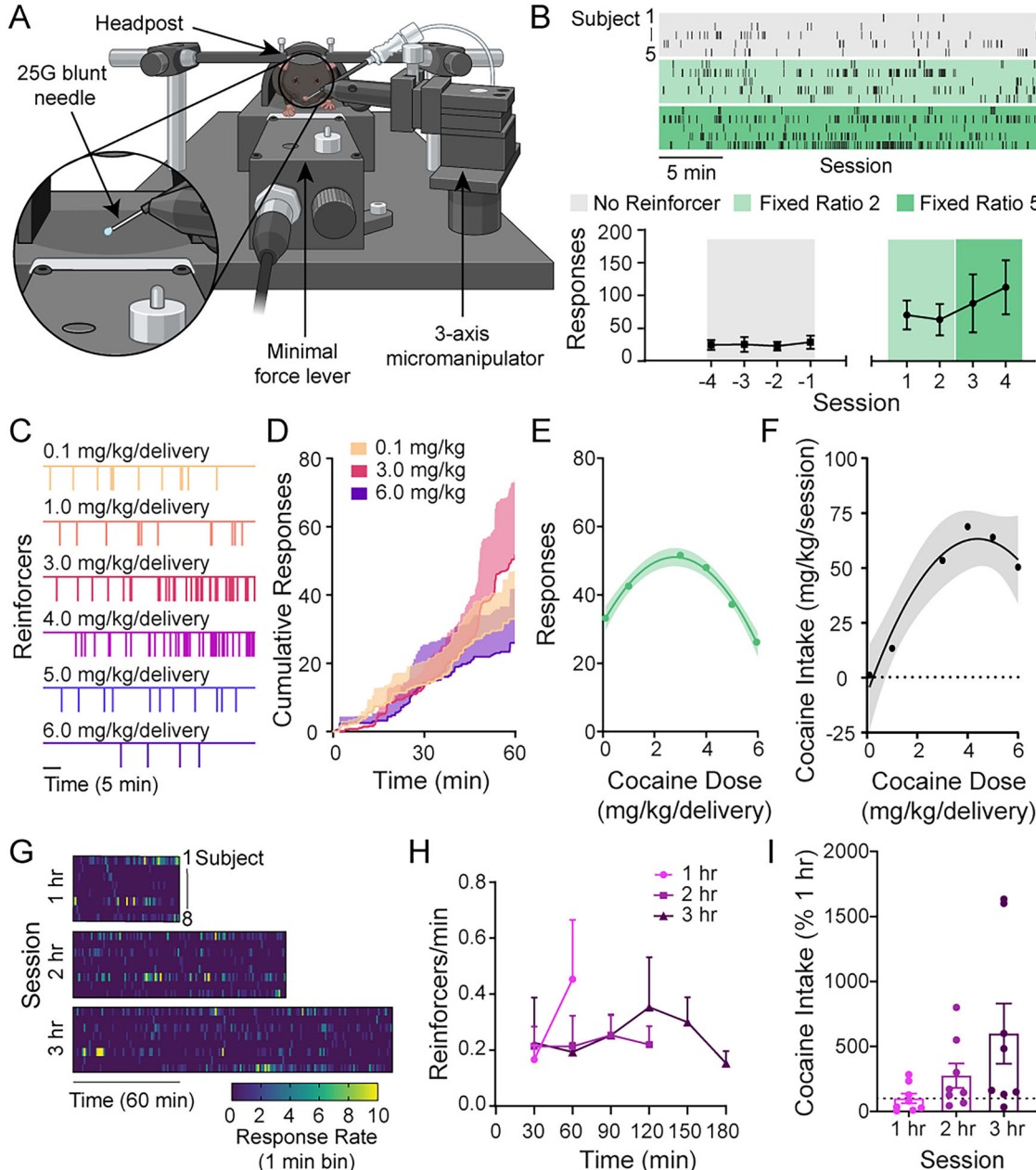

**Fig. 1 | Intranasal cocaine self-administration in head-restrained mice.**
**A** Schematic of experimental apparatus. Mice are head-restrained, and responding on the cocaine-paired lever resulted in the immediate delivery of a 0.5 μL droplet of cocaine solution (dissolved in saline) directly in front of the subject's nostril through a microfluidic delivery system consisting of a blunt needle mounted on a 3-axis miniature micromanipulator. **B** Top: response records from the last session of each session type (no reinforcer, fixed ratio 2, fixed ratio 5). Each row represents response records for an individual animal, with consistent order cross session type. Bottom: mean operant responding per session prior to cocaine availability (responses had no programmed consequence) and when responding is reinforced by 4.0 mg/kg droplets of cocaine under fixed ratio 2 or 5 schedules. Responding increased when reinforced by 4.0 mg/kg cocaine (two-tailed repeated measures one-way ANOVA, dose, $F_{(1.347, 12.12)} = 5.19$, $p = 0.033$). $n = 5$ tested within subject across conditions. **C** Example event records of reinforced responses across increasing doses of cocaine (0.1-6.0 mg/kg/delivery) under a fixed ratio 2 schedule in 1-hour sessions. **D** Averaged

cumulative records of lever responding during low (0.1), middle (3.0), and high (6.0) dose sessions. Responding (second order polynomial, $R^2 = 0.99$) (**E**) and cocaine intake (second order polynomial, $R^2 = 0.96$) (**F**) across the dose-response curve. Error represents 95% confidence interval of the best-fit values. **G** Heatmap of responding across extended-access cocaine self-administration sessions, binned by time (1-minute). Each row is an individual animal, with consistent sorting across sessions. **H** Averaged rate of cocaine reinforcers delivered per minute, binned by time (30-minute), across extended-access sessions. $n = 8$ tested within subject across conditions. Error bars are ± SEM. **I** Cocaine intake increased linearly across extended-access sessions when responding is reinforced by 4.0 mg/kg droplets of cocaine under fixed ratio 2 schedule, normalized to 1 hour session intake (dotted line) (extra sum of squares F test, $F_{(1,22)} = 6.05$, $p = 0.02$ compared to 0). $n = 8$ tested within subject across conditions. Data represent mean ± SEM, with the exception of panels E and F which show 95% confidence interval of the best-fit values. Created in BioRender. Erickson, K. (2025) https://BioRender.com/wvwzyuv.

in place, but responding on the lever had no programmed consequence. After four baseline sessions, cocaine was made available contingent on lever pressing. Cocaine was delivered under a fixed ratio 2 schedule of reinforcement for two sessions, followed by two sessions

of fixed ratio 5. Responding on the cocaine-paired lever increased when reinforced by 4 mg/kg droplets of cocaine (Fig. 1B), demonstrating that cocaine functions as an effective reinforcer under these conditions.

**Repeated nanodroplet insufflation does not cause pulmonary inflammation**

Safe fluid delivery volumes have been established for non-contingent intranasal infusions in rodents[13–15]. 20-30 μL intranasal infusions are well-tolerated in mice, many times larger than used here; however, these methods typically involve a single infusion, and, as such, standards for repeated, sub-microliter infusions have not been established. Therefore, we sought to verify that the volume of fluid ingested by the experimental subjects was tolerable and did not result in aspiration in the lungs. Following repeated intranasal self-administration sessions, direct assessment of the pulmonary tract revealed no fluid infiltrate or markers of inflammation or damage, demonstrating that this configuration is safe for the respiratory system and that droplet volume was sufficiently below levels that produce aspiration (Supplementary Fig. 1).

**Intranasal cocaine drives dose-dependent and sustained intake**

Dose-dependent responding for intravenous cocaine is well-established in humans, monkeys, and rats, but has been difficult to achieve in mice. Additionally, non-mouse models typically employ extended-access self-administration sessions, where high levels of intake induce addiction-like behavior[16,17], but mouse models have been limited to short-access sessions, which do not engender addiction-like changes in cocaine intake[5,6,18,19]. The fact that intravenous cocaine self-administration is particularly difficult in mice and does not produce robust reinforcement has been a major contributing factor to the decline in the utility of the approach, as the use of mice as a model species in neuroscience has become near ubiquitous. Therefore, we aimed to evaluate whether mice would display dose-dependent responding for intranasal cocaine and whether responding would be maintained during extended head-restrained self-administration sessions. First, during daily 1-hour sessions, cocaine was made available under a fixed ratio 2 schedule with increasing doses (0.1 – 6.0 mg/kg/delivery) available across sessions. We observed the quintessential inverted U-shaped relationship between responses relative to cocaine dose, as well as dose-dependent intake (Fig. 1C–F)[20–22]. Following dose-response sessions, cocaine (4.0 mg/kg/delivery) was made available under a fixed ratio 2 schedule for extended 2- and 3-hour sessions. Animals maintained responding throughout both extended access sessions (Fig. 1 G–H) and demonstrated a linear increase in cocaine intake (Fig. 1I).

**Intranasal cocaine self-administration robustly elevates plasma cocaine levels**

To empirically test that the intranasal delivery apparatus allowed subjects to reliably ingest the delivered cocaine as expected, blood was collected immediately following an intranasal self-administration session via submandibular puncture (Fig. 2A). Plasma was analyzed to determine the concentration of cocaine using liquid chromatography-tandem mass spectrometry (LC-MS/MS). Cocaine self-administering mice exhibited elevated plasma cocaine levels compared to controls (Fig. 2B). The observed values (6.3 ± 2.4 μg/mL mean ± SEM, equivalent to 20.6 μM) are greater than what is typically observed following a single intravenous bolus of cocaine or following intravenous self-administration[23–26]. Furthermore, the observed concentrations are comparable to peak blood concentrations reached in humans following intranasal or intravenous administration[27–30]. Additional analysis revealed that the rate of cocaine intake (mg/kg/min) during the self-administration session positively correlated with plasma cocaine concentration (Fig. 2C). Together, these data demonstrate that our intranasal cocaine self-administration protocol 1) effectively delivers cocaine into systemic circulation, 2) results in plasma concentrations proportional to the amount and rate of intake, and 3) drives intake and blood levels relevant for modeling heavy cocaine use.

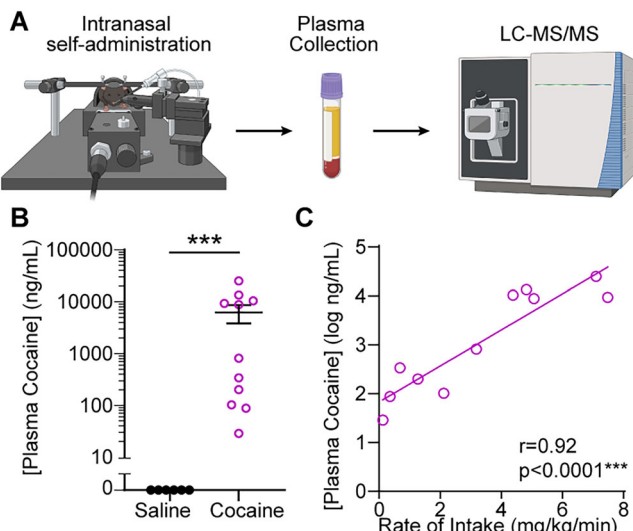

**Fig. 2 | Intranasal cocaine self-administration robustly elevates blood cocaine concentrations as a function of intake. A** Schematic of experimental workflow. Mice underwent an intranasal cocaine self-administration session where cocaine (4 mg/kg) was available under a fixed ratio 1 schedule. To impose structured variance in the amount and timing of intake, the session was terminated when animals met a delivery cap of either 20 ($n = 5$) or 40 ($n = 6$) deliveries, or after 1 hour. Immediately following the session plasma was isolated for later analysis via liquid chromatography-tandem mass spectrometry (LC-MS/MS). **B** Plasma cocaine concentrations (ng/mL) in mice that self-administered cocaine compared to controls. Cocaine-administering mice showed markedly elevated plasma cocaine levels compared to controls (two-tailed Mann-Whitney U test, U = 0, $p = 0.0002$; $n = 5$ control, $n = 11$ cocaine). **C** There was a strong positive correlation between the rate of cocaine intake (mg/kg/min) and log-transformed plasma cocaine concentrations (Pearson's correlation, $r = 0.92$, $p < 0.0001$), indicating that higher rates of self-administration were associated with greater systemic exposure. Data represent mean ± SEM. Created in BioRender. Siciliano, C. (2025) https://BioRender.com/n2j821o.

**Responding is dependent on cocaine delivery**

Next, we sought to verify whether responding in this paradigm was primarily driven by the contingent delivery of cocaine. This is critical, as mice will often lever press in the absence of a contingency, and will respond robustly for cues or other seemingly neutral stimuli[31,32]. To test this, two groups of mice were given access to either intranasal saline (0.5 μL of 0.9% NaCl per delivery) or cocaine (4 mg/kg cocaine dissolved in 0.5 μL of 0.9% NaCl per delivery), and were tested under otherwise identical conditions. Mice were given access in daily three-hour sessions whereby responding was reinforced under a series of fixed ratio schedules (1, 2, 5, and 20) over sessions. Mice receiving cocaine maintained high levels of responding across increasing ratio requirements, whereas the saline group showed minimal engagement (Fig. 3A,B). To examine the temporal patterns of consumption we analyzed inter-delivery intervals across fixed ratio schedules. Cocaine self-administration was associated with rapid and consistent delivery intervals, even under high effort conditions, whereas saline sessions yielded longer and more variable response patterns (Supplementary Fig. 2). These data further support the robust nature of intranasal cocaine reinforcement.

**Intranasal cocaine supports high effort responding**

To determine the reinforcing efficacy of intranasal cocaine, mice were next tested on a progressive ratio schedule. Here, the ratio requirement was increased exponentially for each subsequent infusion (1, 3, 5, 8, etc.), with 4.0 mg/kg of cocaine or 0.5 μL of saline delivered upon

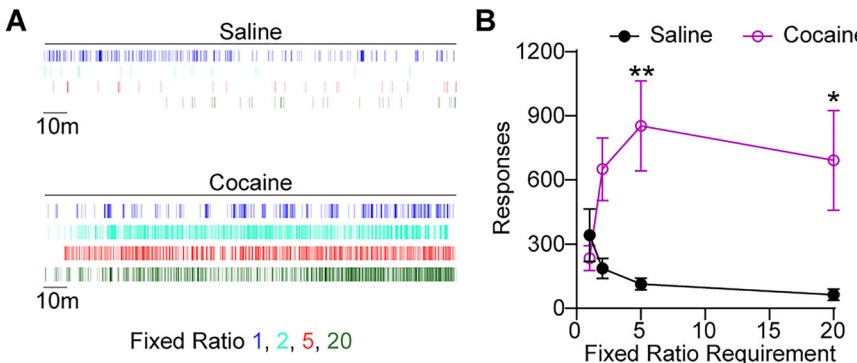

**Fig. 3 | Intranasal cocaine maintains both low and high rates of responding depending on reinforcement schedule. A** Raster plots showing lever presses over time in a representative animal from the saline (top) and cocaine (bottom) groups during fixed ratio self-administration sessions. Each vertical tick mark represents a lever press. Color indicates fixed ratio schedule: fixed ratio 1 (blue), 2 (turquoise), 5 (red), and 20 (green). **B** Operant responding across increasing fixed ratio requirements reinforced by intranasal infusion of saline (0.5 µL of 0.9% NaCl per delivery) or cocaine (4 mg/kg cocaine dissolved in 0.5 µL of 0.9% NaCl per delivery). As expected, high doses of cocaine delivered under a low ratio schedule engendered response rates comparable to saline. However, increasing the ratio requirement resulted in large increases in responding for cocaine while responding for saline decreased. Mice self-administering cocaine displayed higher response rates at fixed ratio 5 and fixed ratio 20 compared to saline controls (two-way ANOVA, drug condition ($F_{(1,47)} = 17.52$, $p = 0.0001$); schedule $F_{(3,47)} = 0.6405$, $p = 0.59$; drug condition x schedule, $F_{(3, 47)} = 3.381$, $p = 0.03$). Sidak's posttest: $*p < 0.05$; $**p < 0.01$ versus saline. Data represent mean ± SEM; $n = 6$ saline, $n = 8$ cocaine.

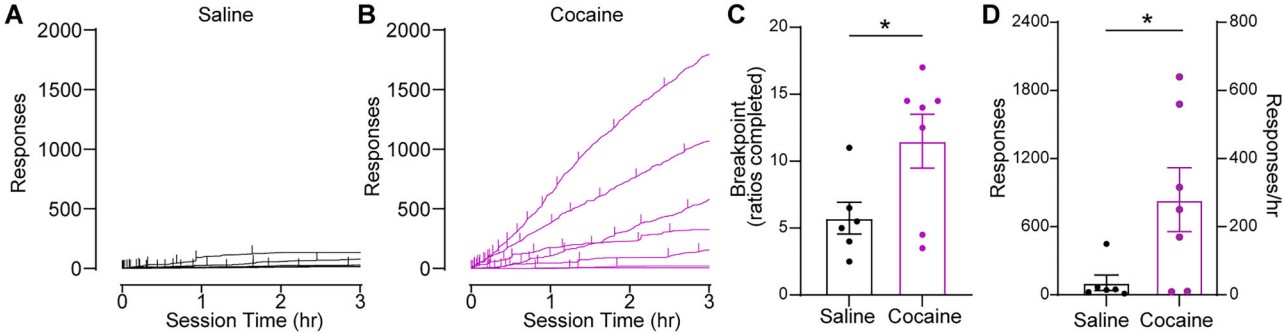

**Fig. 4 | Intranasal cocaine drives high effort responding under a progressive ratio schedule.** Cumulative records of responding from individual mice during a 3-hour progressive ratio session reinforced by delivery of (**A**) saline or (**B**) cocaine (4 mg/kg/delivery). Each line represents an individual animal; vertical ticks indicate a completed ratio / delivery of cocaine or saline. **C** Breakpoint was greater in cocaine-administering mice compared to saline (unpaired t-test, $t_{(11)} = 2.273$, $p = 0.04$). **D** Total number of lever presses was greater in the cocaine group (unpaired t-test, $t_{(11)} = 2.631$, $p = 0.02$). Data represent mean ± SEM; $n = 6$ saline, $n = 7$ cocaine.

completion of each ratio. Cumulative response records (Fig. 4A–B) revealed sustained and escalating engagement through the session in the cocaine group, while responding for saline was minimal and often ceased early. Cocaine-administering animals exhibited higher breakpoints (Fig. 4C) and responses (Fig. 4D) compared to saline self-administering animals. Response rates observed here ($280.7 ± 94.03$ responses/hour for three hours [mean ± SEM]) were considerably higher than prior reports measuring progressive ratio responding for intravenous cocaine in mice[33,34] and are within the range observed for intranasal or intravenous cocaine presented on a progressive ratio schedule in humans[35–37].

### Intranasal self-administration procedure accommodates behavioral economic analysis of cocaine demand

Next, we wanted to evaluate the utility of our task in producing individual differences in cocaine consumption and motivation whereby some individuals are willing to expend a great deal of effort to obtain the drug. To address this, we designed a within-session threshold procedure, which is commonly employed to investigate drug taking and motivation during intravenous cocaine self-administration in humans and rats, but rarely in mouse models. Behavioral economics tasks provide a comprehensive framework for studying the motivational properties of drug and non-drug reinforcers using shared units

of measurement across species; establishing whether intranasal self-administration in mice conforms to the same principles is an important step in determining the veracity of this approach as well as demonstrating its utility in future studies.

Mice were given access to a descending series of 9-unit doses of cocaine ($180 - 2$ µg cocaine/delivery) on a continuous schedule of reinforcement (i.e., fixed ratio 1) over the course of a six-hour session. During discrete 40-minute time bins, the unit price of cocaine, defined as the number of responses required to obtain 1 mg, was increased by decreasing the amount of cocaine per delivery (Fig. 5A). To maintain a preferred blood level of cocaine as the price is increased throughout the session, the animal must increase responding accordingly. Eventually, the price becomes high enough that the subject is no longer willing to expend the effort to maintain their preferred level of the drug, and consumption drops (Fig. 5B). Similar to intravenous self-administration in other species, the interplay between price and consumption for each individual was well-described by an exponential demand function [$R^2 = 0.96 - 0.99$ across animals] (Fig. 5B–C). Measures of demand intensity ($Q_0$), the amount of cocaine consumed at a minimally constrained price, and elasticity ($P_{max}$, alpha), how sensitive the demand for cocaine is relative to changes in the price, are calculated from the best-fit demand curve (Fig. 5B–F). Together, these data support the validity of intranasal cocaine self-administration as a

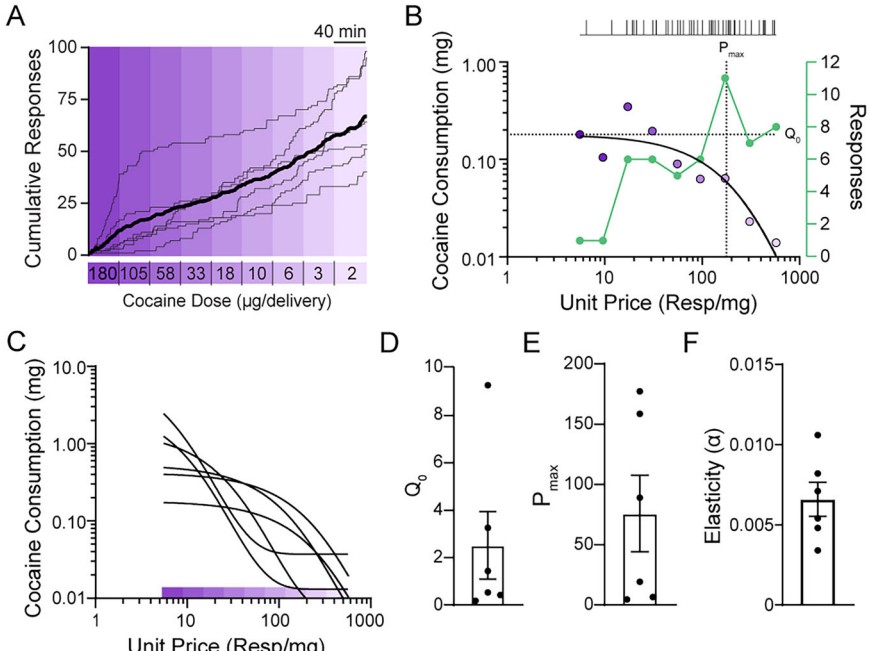

**Fig. 5 | Responding for intranasal cocaine conforms to behavioral economic principles. A** Individual (thin lines) and averaged (thick line) cumulative records of active lever responding with overlay of the behavioral economics task. The unit price, responses required to obtain 1 mg of cocaine, increased across discrete 40-minute time bins by decreasing the amount of cocaine per delivery (180 – 2 µg/delivery). **B** Schematic of behavioral economics analysis parameters. Cocaine consumption per time bin (purple circles) and responses (green circles) were recorded. Values for consumption per bin were curve fit (solid black line) to extract measures of intrinsic value ($Q_0$) and motivation ($P_{max}$) for intranasal cocaine. **C** Demand curves fit to cocaine consumption (mg) per time bin of individual animals. Mean values of $Q_0$ (**D**), $P_{max}$ (**E**), and elasticity (α) (**F**) extracted from demand curves. Data represent mean ± SEM; $n = 6$.

model and the ability of this task to capture individual differences in cocaine consumption and motivation.

In summary, we have introduced a newly developed procedure in mice that allows for operant self-administration of compounds through the intranasal route, and parameterized this model focusing on self-administration of cocaine in male mice. To our knowledge, there are no other documented examples of volitional nasal insufflation of any compound in non-human animals. This procedure does not require any indwelling implants (i.e. catheters, cranial screws, or surgical ports) or surgical expertise, and is compatible with modern techniques that require head immobilization, such as multiphoton imaging and virtual environments. We show that intranasal cocaine drives robust dose- and schedule-dependent responding, akin to rates observed in intravenous self-administration studies in freely moving and head-fixed animals. We anticipate that this procedure can be readily adapted to model intranasal use of a range of other narcotics such as heroin and amphetamines. Future work will be needed to establish this procedure in female mice. Finally, this model provides improved face validity, and putatively improved translational value, given that cocaine is primarily ingested intranasally even among patients with cocaine use disorder[8].

## Methods
### Subjects
Adult, male C57BL/6 J mice (Jackson Laboratory, Bar Harbor, ME; strain #000664) were group-housed (5 per cage) in a reverse 12-hour light-dark cycle (08:00-20:00 lights off). All animals were housed in temperature- and humidity-controlled rooms maintained at $22 \pm 2\,°C$ and 40-60% relative humidity, consistent with institutional standards for rodent care. Animals arrived at the facility at 8 weeks of age and were allowed to acclimate for at least 1 week before any procedures were performed. Food (Picolab 5L0D, LabDiet) and water were available *ad libitum*. All experiments involving the use of animals were in accordance with NIH guidelines and approved by the Vanderbilt Institutional Animal Care and Use Committee.

### Drugs
Cocaine hydrochloride (cocaine) was graciously provided by the National Institute on Drug Abuse (NIDA) through the NIDA Drug Supply Program. Cocaine was dissolved in sterile physiological saline (0.9%, Hospira, Inc., Lake Forest, IL). For all procedures, with the exception of the behavioral economics task (see below), cocaine was dispensed in a volume of 0.5 µL over a 0.5 s period to be insufflated by the subject according to the test protocol and response requirement. Mice were weighed immediately before each session and divided into weight classes based on body weight rounded to the nearest tenth of a gram: 20 g [containing 20-24.9 g mice], 25 g [containing 25-29.9 g mice], or 30 g [containing 30-34.9 g mice]. The concentration of cocaine in the delivery solution was based on the assigned weight class. For example, for a 26 g mouse assigned to the 25-29.9 g weight class requiring a 1.0 mg/kg/delivery cocaine dose, a 50 mg/mL cocaine delivery solution was prepared: $[cocaine] = (\frac{1mg}{0.025kg^{-1}})/\,0.5\mu L^{-1}$. Solutions were prepared every 24-72 hours and stored at $4\,°C$ when not in use.

### Surgical procedure
All surgeries were conducted on mice at least 9 weeks of age using a digital small animal stereotaxic instrument (David Kopf Instruments, Tujunda, CA) under aseptic conditions, and body temperatures were maintained with a heating pad throughout. Animals were anesthetized using isoflurane (5% for induction, 1-2% for maintenance; 1 L/min), positioned in the stereotaxic frame, and ophthalmic ointment was applied to both eyes to prevent corneal desiccation. A small sagittal incision was made along the midline of the skull, and after the surrounding tissue was retracted, a No. 11 scalpel was used to hatch the skull. A custom aluminum anchoring rod (20 ×2 x 2 mm, Shapeways, Livonia, MI) was attached to the stereotax arm and positioned above the frontal part of the skull, parallel to the medio-lateral axis. The skull was allowed to dry completely before affixing the anchoring rod to the aforementioned placement site using Metabond

(C&B Metabond, Parkell). Once dry, the headcap was finalized by encasing it in cranioplastic cement (Ortho-Jet; Lang). At the end of surgery, animals received a warmed subcutaneous injection of keto-profen (5 mg/kg) and Ringer's solution ( ~1 mL), and their body temperatures were maintained using a heating pad until fully recovered from anesthesia. No experiments were performed until a minimum of 1-week post-op.

## Behavioral apparatus

A full list of parts and detailed protocols can be found at https://github.com/Siciliano-Lab/INSA. Briefly, behavioral testing was performed using a custom-built apparatus consisting of an elevated platform and anchor posts that interfaced with the affixed anchoring rod (described under the surgical procedure). Each apparatus was contained within a two-door sound attenuating chamber (18"W x 14"H x 13"D; MedAssociates: CT-SAC-181414-27). A low force lever (AquiNeuro), which required roughly 1.2 g of pressure to engage, was positioned in front of the animal, requiring deliberate forelimb reach to initiate a press.

Cocaine was delivered via a microfluidic system consisting of a 25 G blunt-ended needle mounted on a 3-axis micromanipulator (Thorlabs; DT12XYZ) connected to a high-precision syringe pump (MedAssociates; PHM-210). All sessions were continuously monitored by video captured from a camera that was placed at an angle to the subject to allow for clear observation of the delivery needle. Successful insufflation was confirmed by direct visual observation during each self-administration session. The experimental apparatus allowed precise alignment of the microfluidic delivery needle with the subject's nostril. Upon delivery, animals reliably inhaled the droplet within 1-2 seconds, and failure to do so was rare.

## Behavioral procedures

Mice were given at least 7 days to recover from the previous surgery (headbar implantation surgery) before beginning behavioral experiments. Each session took place in a head-fixation apparatus within a sound attenuating chamber during the active/dark cycle (08:00-20:00 h, only red light was used in the procedure room). Once recovered, mice underwent daily 30 min baseline sessions during which they were placed in the experimental apparatus with the delivery device and lever in place, but no cocaine was available. After four baseline sessions, cocaine (4.0 mg/kg/delivery) was made available contingent on pressing. Cocaine was delivered under a fixed ratio 2 schedule for two sessions, followed by two sessions of fixed ratio 5. Responding on the cocaine-paired lever initiated the immediate delivery of a small droplet of cocaine solution (0.5 μL) directly in front of the subject's nostril (infused over 0.5 s, 1.8 RPM). Each delivery was paired with a tone and light cue. Each reinforced lever press also resulted in a 5 s timeout period during which further responding on the lever was recorded but not reinforced. Cocaine solution was delivered using a high-resolution syringe pump (MED Associates, PHM-210).

**Dose-response curve.** Following habituation, animals underwent six 1 hr sessions of intranasal cocaine self-administration on a fixed ratio 2 schedule of reinforcement using an increasing dose design (0.1, 1.0, 3.0, 4.0, 5.0, and 6.0 mg/kg/delivery, infused over 0.5 s). Each delivery was paired with a tone and light cue, followed by a 5 s timeout period during which responses were recorded but did not result in a cocaine delivery or cues.

**Extended-access self-administration sessions.** Mice were given access to 4.0 mg/kg cocaine under a fixed ratio 2 schedule of reinforcement over the course of 2 or 3 hr sessions. Task parameters matched the dose-response curve, with the exception of a 10 s timeout period following each reinforced lever press.

**Behavioral Economics.** Mice were given access to a descending series of 9-unit doses of cocaine (180, 105, 58, 33, 18, 10, 6, 3, and 2 μg cocaine per delivery) on a continuous, fixed ratio 1 schedule of reinforcement across a 6 hr session. Cocaine concentration (360 mg/mL) and pump speed (0.2 RPM) were held constant, and the amount of cocaine delivered per infusion was modulated throughout the session by decreasing the pump duration (4.39, 2.55, 1.42, 0.79, 0.44, 0.25, 0.14, 0.08, 0.04 seconds per delivery). Accordingly, the cocaine per delivery values listed above thus corresponded to 495, 287, 160, 89, 49, 29, 16, 9, and 5 nL of cocaine solution per delivery. Each dose was available for 40 min, with each bin presented consecutively across the 6 hr session. Prior within-session intravenous cocaine self-administration procedures have utilized 10 min time bins based on the half-life of intravenous cocaine; the intranasal route of administration results in a slower onset and longer duration of action compared to intravenous administration. We therefore adjusted the bin size to 40 min epochs based on the fold difference in clearance[10,38]. Timeout periods, where presses were not reinforced, occurred only during each infusion. Completion of the procedure generated a within-session demand curve (Fig. 2).

The demand curves were fit using the equation $\log Q = \log Q_0 + k \left( e^{-a \times (Q_0 \times C)} - 1 \right)$ to derive $Q_0$, $P_{max}$, and $\alpha$, as previously described[39–42]. $Q_0$ is the consumption as price approaches zero, $P_{max}$ is the first unit price point at which the first derivative point slope of the function equals −1, and $\alpha$ is the rate of change in elasticity as price increases along the demand curve.

**Fixed ratio cocaine self-administration with rolling dose cap.** Mice were trained to self-administer cocaine (4 mg/kg/delivery) under escalating fixed ratio schedules of reinforcement. Animals underwent daily 3 hour self-administration sessions. Mice were initially trained on a fixed ratio 1 schedule for 3 days, followed by 3 days of fixed ratio 2, 3 days of fixed ratio 5, and concluded with 2 days of fixed ratio 20. Across all schedules, when the required number of presses was reached, the task evaluated whether the animal exceeded a rolling cap on total cocaine intake. This rolling cap constrained cocaine exposure to a user-defined dose limit (40 mg/kg) within a moving time window (60 min). The maximum number of infusions permitted per window (Q) was calculated at the session start (Q = cap dose / dose per delivery) and rounded to the nearest whole number. If fewer than Q infusions had been delivered in the most recent time window, a 0.5 μL cocaine or saline droplet was delivered, paired with a compound audiovisual cue. If the rolling cap had been reached, the infusion was withheld and the ratio counter reset without cue delivery. Following each earned or withheld cocaine delivery, a 20-second timeout period was imposed, during which additional lever presses were recorded but did not advance the ratio counter.

**Progressive ratio.** Mice were trained to respond on the lever for intranasal delivery of cocaine (4 mg/kg/delivery) under a progressive ratio schedule of reinforcement, similar to previously described intravenous self-administration studies[43]. The response requirement for each infusion followed a non-linear progression according to the following: 1, 3, 5, 8, 11, 16, 23, 31, 41, 55, 72, 94, 123, 160, 207, 267, 345, 444, 572, and 736. Following each earned or withheld cocaine delivery, a 20-second timeout period was imposed, during which additional lever presses were recorded but did not advance the ratio counter. Sessions continued until the session time limit of 3 hours was reached. Breakpoint was defined as the number of ratios completed (i.e. number of deliveries).

## Histopathology

At the conclusion of the study, a subset of mice were examined for any potential inflammation of the lungs and airways. Mice were humanely euthanized via $CO_2$ inhalation in accordance with the institutional

IACUC protocol. A complete set of tissues was collected into 10% neutral buffered formalin and fixed for approximately 72 hours. Fixed tissues were routinely processed, embedded in paraffin, sectioned at 5 μm, and stained with hematoxylin and eosin. A board-certified veterinary pathologist (K.N.G.C.) reviewed all the slides (Figure S1).

### Plasma cocaine concentrations
**Fixed ratio cocaine self-administration.** Mice underwent daily self-administration session where cocaine (4 mg/kg/delivery) was available under a fixed ratio 1 schedule. Responding on the lever initiated the immediate delivery of a small droplet of cocaine solution (0.5 μL) directly in front of the subject's nostril (infused over 0.5 s, 1.8 RPM). Each delivery was paired with a tone (12 kHz) and light cue. Reinforced responses initiated a 20 s timeout during which further lever presses were recorded but not reinforced. Two sessions were conducted on separate days: one with a maximum of 20 deliveries, and a second with a 40-delivery cap. Each session lasted up to 1 hr or until the delivery cap was reached.

**Sample collection.** Immediately following the self-administration session, blood samples (~100 μL) were obtained via submandibular puncture using a sterile lancet (5.5 mm, Goldenrod). Samples were collected via capillaries into microtubes pre-coated with EDTA to prevent coagulation and were stored briefly on ice. Blood was centrifuged promptly at 2000 x g for 10 minutes at 4 °C to separate plasma. The plasma fraction was transferred into clean tubes and stored at −80 °C until further analysis via liquid chromatography-tandem mass spectrometry.

**Liquid chromatography-tandem mass spectrometry.** Plasma samples (50 μL) were prepared by combining 25 μL of experimental plasma with 25 μL of blank mouse plasma if sample volume was limited. Each sample received 5 μL internal standard solution (1 μg/mL), followed by protein precipitation with 200 μL acetonitrile. Samples were vortexed, centrifuged at 10,000 rpm for 10 min at 4 °C, and the supernatant was transferred to fresh tubes and dried under nitrogen gas. Residues were reconstituted in 100 μL of 0.1% formic acid in water.

Prepared samples were analyzed via liquid chromatography-tandem mass spectrometry (LC-MS/MS) using a Thermo TSQ Quantum Ultra AM mass spectrometer coupled to a Waters Acquity HPLC system. Chromatographic separation was performed on an Agilent Poroshell 120 EC-C18 column (2.5 μm, 3 × 50 mm) with a mobile phase composed of solvent A (0.1% formic acid in water) and solvent B (0.1% formic acid in acetonitrile). A gradient elution was employed at a flow rate of 0.3 mL/min: initial conditions of 5% B were held for 1 min, increased linearly to 95% B over 3.5 min, held at 95% for an additional 1 min, and then returned to initial conditions with re-equilibration for 1.5 min. Data acquisition and analysis were conducted with LC-Quan software (Thermo Scientific).

### Statistics
Statistical analyses were performed using GraphPad Prism (V10). Comparisons across three or more variables were made using one-way, repeated-measures ANOVAs (followed by Šídák's multiple comparisons when planned comparisons were made or interactions were detected). For correlation analyses, Pearson's correlation coefficient was used to test relationships between continuous variables. All tests were two-tailed and $p$ values < 0.05 were considered to be statistically significant.

### Reporting summary
Further information on research design is available in the Nature Portfolio Reporting Summary linked to this article.

## Data availability
Source data for all figures are provided with this paper in the Source Data file. Source data are provided with this paper.

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

## Acknowledgements

This work was supported by NIH grants R00 DA045103 (NIDA), R01 AA030115 (NIAAA), U01 AA029971 (NIAAA), P60 AA031124 (NIAAA), the Alkermes Pathways Research Award, the Brain Research Foundation, the Whitehall Foundation, the W.M. Keck Foundation, and the Stanley Cohen Innovation Fund (C.A.S.). K.R.E and Z.Z.F. were supported by NIH fellowships F31 DA056202 (NIDA) and F31 DA056196 (NIDA). We acknowledge the Translational Pathology Shared Resource supported by NCI/NIH Cancer Center Support Grant P30 CA068485.

## Author contributions

Conceptualization: K.R.E., C.A.S.; Data curation: K.R.E., Y.Q., H.E.B., J.D.K., K.N.G-C., C.A.S.; Formal analysis: K.R.E., Z.Z.F., J.D.K., C.A.S.; Funding acquisition: K.N.G-C., C.A.S.; Investigation: K.R.E., Y.Q., H.E.B., K.S., J.D.K., J.J.L., K.N.G-C., C.A.S.; Methodology: K.R.E., E.Y.K., C.A.S.; Project administration: C.A.S.; Resources: E.Y.K., C.A.S.; Software: Z.Z.F., E.Y.K., C.A.S.; Supervision: K.R.E., C.A.S.; Validation: K.R.E., Y.Q., H.E.B., C.A.S.; Visualization: K.R.E., Z.Z.F., K.N.G-C., C.A.S.; Writing – original draft: K.R.E., C.A.S.; Writing – review & editing: K.R.E., Y.Q., Z.Z.F., H.E.B., K.S., J.D.K., J.J.L., K.N.G-C., E.Y.K., C.A.S.

## Competing interests

The authors declare no competing interests.
