## [Transparent Peer Review file · Nature Communications]

Intranasal cocaine self-administration in mice

Corresponding Author: Dr Cody Siciliano

Version 0:

Reviewer comments:

Reviewer #1

(Remarks to the Author)

This is an exciting, tool-developing study, in which the authors optimized and validated intranasal delivery of cocaine in mice through a head-restrained set up. The data provided are solid and convincing. This setup is expected to be extremely useful for numerous labs (including mine) in the field of SUD. I am highly enthusiastic, but ask two experiments:

- 1) To make this procedure useful, it needs to show that cocaine administered through this route actually gets into the blood and pushes the blood level comparable to IVSA.
- 2) A badly needed feature of this procedure is that cocaine, but not saline (or other controls), establishes a stable reinforcement. The 0.1 mg/kg experiment provides some clues, but can be interpreted in many different ways. It appears that saline alone is a reasonable control to do.

Reviewer #2

(Remarks to the Author)

This report described the development and initial validation of a mice intranasal cocaine self-administration procedure. Mice with fixed head were given access to cocaine droplets and lever-pressing led to the injection of a small droplet in the nose. Positive reinforcing effect of cocaine was examined by increasing lever-pressing requirement, varying cocaine concentration and changing session time. Behavioral outputs suggest cocaine as a positive reinforcer in this setup. It was concluded that this was the first known intranasal cocaine self-administration setup and may have broad use in modern neuroscience research.

I have the following concerns.

Major concerns:

While interesting, the impact of the setup may not be as significant as authors claimed. Since the first report of nonhuman primate self-administration study by Spragg in 1940 and later in rats by Weeks in 1962, numerous self-administration apparatus were designed and reported, including but not limited to intravenous, oral, intramuscular, intragastric, insufflation and smoked drug deliveries across large class of drugs of abuse. Animal species used in drug self-administration studies included chimpanzees, rhesus monkeys, squirrel monkeys, marmosets, pigs, dogs, rats and mice etc. What kind of self-administration setup should be used depends on several factors, such as the face validity (how human drug users do the drug?), technical challenges, and drug class. This report demonstrated that mice can self-administer cocaine via intranasal route which is interesting, but how generalizable is this setup to other drugs of abuse? What is the blood cocaine level during and after a cocaine self-administration session? Are the mice really working for cocaine? Saline substitution and extinction should be tested.

Reviewer #3

(Remarks to the Author)

Noteworthy Results

The authors present a novel method for intranasal cocaine self-administration in head-restrained mice, which they claim is

the first documented example of volitional nasal insufflation in non-human animals. The method shows robust, dose-dependent responding similar to traditional intravenous self-administration. The technique eliminates the need for complex surgical procedures (like jugular catheterization) that have high failure rates. Animals demonstrated expected behavioral economic principles, with demand curves matching those seen in other species. The method allowed for extended access sessions, which has been difficult to achieve in mouse models previously. In general, this is a very significant manuscript that has the potential to advance the field by eliminating surgical attrition (potentially...see below) and opens the door to imaging/modern technology usage during self-administration.

Significance and Originality

The work appears highly significant as it addresses several major technical limitations in current addiction research methods:

Reduces surgical complexity and animal attrition

Better matches human routes of administration

Enables integration with modern neuroscience techniques requiring head fixation

The approach is original - no prior studies have demonstrated voluntary intranasal drug administration in animals. The authors appropriately cite and build upon relevant literature in the field.

Support for Conclusions

The data generally support the main claims:

Clear demonstration of dose-dependent responding

Evidence of safe administration (no respiratory complications)

Behavioral economic analyses showing expected patterns

Area needing clarification/more detail:

Blood level analysis is mentioned, but data are not provided

Some additional evidence or discussion of the following would strengthen the work:

Direct comparison to traditional IV self-administration methods

Pharmacokinetic data comparing intranasal vs IV routes

More detailed safety/toxicology data over longer time periods

Are the lungs the only area of concern with intranasal infusion, or might septal perforation be an issue?

Inclusion of females would be helpful to confirm they are equally safe/effective in both sex.

Methodology and Reproducibility

Strengths:

Clear description of apparatus and basic procedures

Multiple behavioral measures validate the approach

Appropriate controls and behavioral economic analyses

Areas needing clarification/More detail needed on:

Head-fixation surgical procedure

Precise composition of cocaine solution

Criteria for excluding animals/sessions

Statistical power calculations

Method for confirming successful insufflation

Initial response curves after the four baseline sessions are complete

Details on how animals were run; 5 simultaneously? Same time of day? Given small n (5), perhaps there is a limitation of the number of animals that can be run concurrently

Github link in supplement delivers 404 error

Temperature and humidity range in housing and experimental rooms

Data Analysis:

Statistical analyses appear appropriate

Good use of established behavioral economic frameworks

Clear presentation of individual and group data

Some minor concerns:

Sample sizes could be larger for some experiments

More detail needed on handling of outliers/exclusions

Additional statistical tests for some comparisons would be helpful

Overall Assessment:

This appears to be a significant methodological advance that could have a broad impact on addiction research. The core findings are well-supported, though some additional details would strengthen the work. The manuscript would benefit from:

More detailed methods

Additional control experiments

Direct comparisons to traditional methods

Expanded discussion of limitations

These revisions would likely be achievable without additional major experiments. The work appears suitable for publication after addressing these points, and would be a notable advance for the field.

Reviewer #4

(Remarks to the Author)

Version 1:

Reviewer comments:

Reviewer #1

(Remarks to the Author)

the authors did a very good job revising the manuscript. I do not have additional concerns.

Reviewer #2

(Remarks to the Author)

I thank the authors to provide a thorough revision including the rebuttal (which I accept) and additional experiments. I have no more concerns.

We would like to thank the reviewers and editors for their time and constructive comments on our manuscript. At the request of the reviewers, we have added multiple new experiments including testing across multiple fixed ratio schedules, responding under a progressive ratio schedule of reinforcement, saline self-administration controls, and liquid chromatography-tandem mass spectrometry analysis of plasma cocaine concentrations. Further, we have expanded our discussion and comparison with the prior literature and clarified several methodological points. We are grateful for the reviewers' insightful suggestions and we believe these changes have considerably strengthened the impact, rigor, and clarity of the manuscript.

Responses to Individual Reviewer Comments:

Please find point-by-point responses to each reviewers' comments, colored coded by reviewer:

Reviewer #1

Reviewer #2

Reviewer #3

Reviewer #4

Our point-by-point responses are underlined. For the reviewers' convenience, we have copied revised passages from the manuscript below their associated comments; *text taken directly from the manuscript is denoted by italics*. In the manuscript file, revised and new passages are denoted in **blue**

Reviewer #1:

Comment 0: This is an exciting, tool-developing study, in which the authors optimized and validated intranasal delivery of cocaine in mice through a head-restrained set up. The data provided are solid and convincing. This setup is expected to be extremely useful for numerous labs (including mine) in the field of SUD.

Response 0: We thank the reviewer for highlighting the importance of the work, and for the excellent experimental suggestions.

Comment 1: I am highly enthusiastic, but ask two experiments: To make this procedure useful, it needs to show that cocaine administered through this route actually gets into the blood and pushes the blood level comparable to IVSA.

Response 1: We agree this information would be useful and have now added additional experiments showing plasma cocaine concentrations following an intranasal self-administration session. We find that intranasal self-administration produces robust elevations in plasma cocaine concentrations within a one-hour session ($6.3 \pm 2.4 \mu\text{g} / \text{mL}$ mean \pm SEM, equivalent to $20.6 \mu\text{M}$). We were not able to find prior literature reporting blood levels resulting from intravenous self-administration of cocaine in mice, but compared to intravenous self-administration in rats, these values are at the high end of the range that has been typically reported (Bystrowska et al., 2012; Lau & Sun, 2002). Furthermore, these values are comparable to peak blood concentrations reached in humans following intranasal or intravenous administration (Barnett et al., 1981; Isenschmid et al., 1992; McGrath et al., 2020; Van Dyke et al., 1976). Further, the plasma concentrations observed here were highly correlated with rate of intake during the session ($r=0.92$, **Figure 2**). Together, these data demonstrate that our intranasal cocaine self-administration protocol 1) effectively delivers cocaine into systemic circulation, 2) results in plasma cocaine concentrations proportional to the amount and rate of intake, and 3) drives intake and blood levels relevant for modeling heavy cocaine use.

Comment 2: A badly needed feature of this procedure is that cocaine, but not saline (or other controls), establishes a stable reinforcement. The 0.1 mg/kg experiment provides some clues, but can be interpreted in many different ways. It appears that saline alone is a reasonable control to do.

Response 2: While the original submission did include a non-reinforced condition where the animals had access to the lever but the operant was not in effect, we agree that a fully matched control, where cues and saline are delivered under an identical schedule, is critical. We have now included additional experiments comparing saline vs cocaine self-administration, tested under identical conditions, across a range of fixed ratio schedules (**Figure 3**) as well as on a progressive ratio (**Figure 4**). Both experiments support the notions that 1) cocaine functions as a powerful reinforcer when presented using our intranasal self-administration protocol and 2) that many of the characteristics of intravenous cocaine self-administration in freely moving rats apply to intranasal self-administration in head-restrained mice.

Reviewer #2

This report described the development and initial validation of a mice intranasal cocaine self-administration procedure. Mice with fixed head were given access to cocaine droplets and lever-pressing led to the injection of a small droplet in the nose. Positive reinforcing effect of cocaine was examined by increasing lever-pressing requirement, varying cocaine concentration and changing session time. Behavioral outputs suggest cocaine as a positive reinforcer in this setup. It was concluded that this was the first known intranasal cocaine self-administration setup and may have broad use in modern neuroscience research.

I have the following concerns.

Major concerns:

- Comment 1.0:** While interesting, the impact of the setup may not be as significant as authors claimed. Since the first report of nonhuman primate self-administration study by Spragg in 1940 and later in rats by Weeks in 1962, numerous self-administration apparatus were designed and reported, including but not limited to intravenous, oral, intramuscular, intragastric, insufflation and smoked drug deliveries across large class of drugs of abuse. Animal species used in drug self-administration studies included chimpanzees, rhesus monkeys, squirrel monkeys, marmosets, pigs, dogs, rats and mice etc. What kind of self-administration setup should be used depends on several factors, such as the face validity (how human drug users do the drug?), technical challenges, and drug class. **Comment 1.1:** This report demonstrated that mice can self-administer cocaine via intranasal route which is interesting, but how generalizable is this setup to other drugs of abuse? **Comment 1.2:** What is the blood cocaine level during and after a cocaine self-administration session? **Comment 1.3:** Are the mice really working for cocaine? Saline substitution and extinction should be tested.

Response 1.0: We agree with the reviewer that multiple factors, including face validity, are critical in selecting a self-administration model – we did not intend to imply that our method should replace intravenous models across the board nor do we think that a one-size-fits-all approach would be appropriate for model selection in general. Regarding the potential impact, while it is true that many self-administration paradigms have been performed, it is worth noting that the technique has seen a steady decline in usage in the field over the last decade (Rebuttal Figure 1), roughly corresponding to the proliferation of techniques which require tethering and/or head-fixation in neuroscience. Further, in the early literature, drug self-administration was utilized, to great effect, by basic researchers studying motivation outside of the context of clinically relevant addiction studies; this has become much less common in the modern literature, likely due to the difficulty of the approach. In addition to the impact for the addiction field, we hope that this work will re-open the door to leveraging the motivational properties of drugs for the behavioral neuroscience community at large.

In the original submission, we focused on the relative ease of this method compared to commonly used approaches such as intravenous cocaine self-administration. We feel these advantages are substantial but failed to clearly explain that this approach also provides greatly improved face validity in several scenarios. Indeed, intranasal drug use is the most common route of administration during initiation of narcotic use across several compounds. After chronic use of most narcotics, heavy users often turn to other routes such as intravenous or smoked. However, in the case of cocaine, even among heavy users, the intranasal route is more common than intravenous (Sanvisens et al., 2021). This unique aspect of cocaine, whereby the intranasal route is widely favored among recreational and heavy users, was a major motivating factor for us in developing this approach.

Rebuttal Figure 1. **Relative frequency of self-administration publications in rodents.** Rolling five year average of the number of PubMed hits for search term: ("Mice"[Mesh] OR ("Rats"[Mesh])) AND (administration, self[MeSH Terms]) divided by the number of hits for: ("Mice"[Mesh] OR ("Rats"[Mesh])).

Together, in addition to greatly improved ease of implementation, this approach provides increased face validity for the initiation of drug use across classes, and for cocaine use across stages of addiction.

Response 1.1: We anticipate that this methodology will be readily adaptable for use with other narcotics such as opioids and amphetamines. For the reasons explained in Response 1.0, we have chosen to focus on cocaine in the current manuscript, and have now clarified this rationale and expanded on the possibility of adapting for other compounds in the manuscript.

Response 1.2: We agree this information would be useful and have now added additional experiments showing plasma cocaine concentrations following an intranasal self-administration session. We find that intranasal self-administration produces robust elevations in plasma cocaine concentrations within a one-hour session ($6.3 \pm 2.4 \mu\text{g} / \text{mL}$ mean \pm SEM, equivalent to $20.6 \mu\text{M}$). These values are at the high end of the range that has been typically reported for intravenous self-administration in rats (Bystrowska et al., 2012; Lau & Sun, 2002), and are comparable to peak blood concentrations reached in humans following intranasal or intravenous administration (Barnett et al., 1981; Isenschmid et al., 1992; McGrath et al., 2020; Van Dyke et al., 1976). Further, the plasma concentrations observed here were highly correlated with rate of intake during the session ($r=0.92$, **Figure 2**). Together, these data demonstrate that our intranasal cocaine self-administration protocol 1) effectively delivers cocaine into systemic circulation, 2) results in plasma cocaine concentrations proportional to the amount and rate of intake, and 3) drives intake and blood levels relevant for modeling heavy cocaine use.

Response 1.3: We have now included multiple additional experiments to test the degree to which responding is cocaine-dependent. First, we have now included a comparison of saline vs cocaine-self-administration across multiple fixed-ratio requirements, which demonstrates that mice respond more for cocaine, but not saline, as the ratio requirement is increased (**Figure 3**). Along these same lines, we have now included additional analysis of response periodicity in cocaine (**Figure S2**) vs saline self-administering animals, as well as cocaine self-administration reinforced under a progressive ratio schedule (**Figure 4**). Throughout, all of these data are consistent with the claim that cocaine is functioning as a powerful reinforcer when presented using our intranasal self-administration protocol.

Reviewer #3

Comment 0: Noteworthy Results

The authors present a novel method for intranasal cocaine self-administration in head-restrained mice, which they claim is the first documented example of volitional nasal insufflation in non-human animals. The method shows robust, dose-dependent responding similar to traditional intravenous self-administration. The technique eliminates the need for complex surgical procedures (like jugular catheterization) that have high failure rates. Animals demonstrated expected behavioral economic principles, with demand curves matching those seen in other species. The method allowed for extended access sessions, which has been difficult to achieve in mouse models previously. In general, this is a very significant manuscript that has the potential to advance the field by eliminating surgical attrition (potentially...see below) and opens the door to imaging/modern technology usage during self-administration.

Significance and Originality

The work appears highly significant as it addresses several major technical limitations in current addiction research methods:

Reduces surgical complexity and animal attrition

Better matches human routes of administration

Enables integration with modern neuroscience techniques requiring head fixation

The approach is original - no prior studies have demonstrated voluntary intranasal drug administration in animals. The authors appropriately cite and build upon relevant literature in the field.

Support for Conclusions

The data generally support the main claims:

Clear demonstration of dose-dependent responding

Evidence of safe administration (no respiratory complications)

Behavioral economic analyses showing expected patterns

Response 0: We thank the reviewer for their thoughtful and supportive comments. We agree that the integration of this model with modern tools offers a substantial promise for advancing mechanistic studies of addiction, and we are grateful for the reviewers recognition of that potential.

Comment 1: Area needing clarification/more detail:

Blood level analysis is mentioned, but data are not provided

Response 1: We agree this information would be useful and have now added additional experiments showing plasma cocaine concentrations following an intranasal self-administration session. We find that intranasal self-administration produces robust elevations in plasma cocaine concentrations within a one-hour session ($6.3 \pm 2.4 \mu\text{g} / \text{mL}$ mean \pm SEM, equivalent to $20.6 \mu\text{M}$). These values are at the high end of the range that has been typically reported for intravenous self-administration in rats (Bystrowska et al., 2012; Lau & Sun, 2002), and are comparable to peak blood concentrations reached in humans following intranasal or intravenous administration (Barnett et al., 1981; Isenschmid et al., 1992; McGrath et al., 2020; Van Dyke et al., 1976). Further, the plasma concentrations observed here were highly correlated with rate of intake during the session (**$r=0.92$, Figure 2**). Together, these data demonstrate that our intranasal cocaine self-administration protocol 1) effectively delivers cocaine into systemic circulation, 2) results in plasma cocaine concentrations proportional to the amount and rate of intake, and 3) drives intake and blood levels relevant for modeling heavy cocaine use.

Comment 2: Some additional evidence or discussion of the following would strengthen the work:

Direct comparison to traditional IV self-administration methods

Response 2: We have expanded our discussion of how our results with intranasal self-administration compare to intravenous self-administration in mice.

Comment 3: Pharmacokinetic data comparing intranasal vs IV routes

Response 3: Though intranasal cocaine self-administration in animals has not been reported in the prior

literature, pharmacodynamics between intranasal and intravenous cocaine have been extensively compared following non-contingent administration in rodents and self-administration in humans. Intravenous (IV) cocaine produces an immediate rise in plasma and brain concentrations, with peak levels typically achieved within seconds due to direct delivery into systemic circulation and rapid brain penetration (Javaid et al., 1978; Volkow et al., 2000). In contrast, intranasal administration leads to a slower absorption profile, with peak plasma concentrations occurring approximately 5-10 minutes post-administration, reflecting transnasal mucosal absorption (Cone, 1995; Jeffcoat et al., 1989). Although intranasal administration results in lower initial peak concentrations relative to IV, it still produces substantial systemic exposure, with reported bioavailability forming from 60 to 80% (Bravo, 2022; Jeffcoat et al., 1989). Despite differences in absorption rate and peak concentration, the elimination half-life of cocaine is similar across routes, typically ranging from 40 to 90 minutes (Bravo, 2022; Cone, 1995).

Additionally, we have now added additional experiments showing blood cocaine concentrations following an intranasal self-administration session (see Response to Comment 1).

Comment 4: More detailed safety/toxicology data over longer time periods

Are the lungs the only area of concern with intranasal infusion, or might septal perforation be an issue?

Response 4: Regarding safety, our prior concern was pulmonary injury given that the cocaine is dissolved in solution – any other safety concerns from the cocaine itself would be an aspect of the model worth investigating, as it would be relevant for human cocaine use, rather than an issue with the model. Further, many of the deleterious topical effects of cocaine use are driven by adulterants in street cocaine and secondary effects such as infection (Faelens et al., 2021; Shannon, 1988). Finally, it is worth noting that mice do not have separate nasal cavities, they have an opening called the ‘window of the nasal septum’ that connects the two cavities, and they are therefore considered as a single cavity in regard to intranasal fluid delivery (Alvites et al., 2018; Navarro et al., 2017).

Comment 5: Inclusion of females would be helpful to confirm they are equally safe/effective in both sex.

Response 5: We agree that ultimately it is critical to test these questions in both sexes. We and others have recently called for greater emphasis on sex x environment interactions rather than claims of sexual dimorphism in neuroscience studies (Brown et al., 2023; Lewis et al., 2021; Zachry et al., 2019); accordingly, we find that investigating within each sex in parallel, rather than attempts to parse sex differences *per se*, often produces more clear results when discovery of mechanistic reasons for dimorphisms is not the explicit goal (Brown et al., 2023). We have now added a discussion point regarding the need for similar experiments in female subjects. We hope that this is not perceived as an attempt to sidestep the issue – we sincerely plan to continue to pursue parallel investigations in both sexes, and have a track record of doing so in prior work (Brown et al., 2023; Nolan et al., 2025; Siciliano et al., 2015, 2016, 2019); if the reviewers or editors still feel that the manuscript would be best served by combining these efforts here, we can do so.

Comment 6: Methodology and Reproducibility

Strengths:

- Clear description of apparatus and basic procedures
- Multiple behavioral measures validate the approach
- Appropriate controls and behavioral economic analyses

Areas needing clarification/More detail needed on:

Head-fixation surgical procedure

Response 6: This has been clarified in the methods section, as detailed below:

All surgeries were conducted on mice at least 9 weeks of age using a digital small animal stereotaxic instrument (David Kopf Instruments, Tujunga, CA) under aseptic conditions, and body temperatures were maintained with a heating pad throughout. Animals were anesthetized using isoflurane (5% for induction, 1-2% for maintenance; 1 L/min), positioned in the stereotaxic frame, and ophthalmic ointment was applied to both

eyes to prevent corneal desiccation. A small sagittal incision was made along the midline of the skull, and after the surrounding tissue was retracted and a No. 11 scalpel was used to hatch the skull. A custom aluminum anchoring rod (20 x 2 x 2 mm, Shapeways, Livonia, MI) was attached to the stereotax arm and positioned above the frontal part of the skull, parallel to the medio-lateral axis. The skull was allowed to dry completely before affixing the anchoring rod to the aforementioned placement site using Metabond (C&B Metabond, Parkell). Once dry, the headcap was finalized by encasing it in cranioplastic cement (Ortho-Jet; Lang). At the end of surgery, animals received a warmed subcutaneous injection of ketoprofen (5 mg/kg) and Ringer's solution (~1 mL), and their body temperatures were maintained using a heating pad until fully recovered from anesthesia. No experiments were performed until a minimum of 1-week post-op.

Comment 7: Precise composition of cocaine solution

Response 7: This has been clarified in the methods section, as detailed below:

Cocaine hydrochloride (cocaine) was graciously provided by the National Institute on Drug Abuse (NIDA), through the NIDA Drug Supply Program. Cocaine was dissolved in sterile physiological saline (0.9%, Hospira, Inc., Lake Forest, IL). For all procedures, with the exception of the behavioral economics task (see below), cocaine was dispensed in a volume of 0.5 μL over a 0.5 s period to be insufflated by the subject according to the test protocol and response requirement. Mice were weighed immediately before each session and divided into weight classes, based on body weight rounded to the tenth of a gram: 20 g [containing 20-24.9 g mice], 25 g [containing 25-29.9 g mice], or 30 g [containing 30-34.9 g mice]. The concentration of cocaine in the delivery solution was based on the assigned weight class. For example, for a 26 g mouse assigned to the 25-29.9 g weight class requiring a 1.0 mg/kg/delivery cocaine dose, a 50 mg/mL cocaine delivery solution was prepared: $[\text{cocaine}] = \left(\frac{1 \text{ mg}}{0.025 \text{ kg}}\right) / 0.5 \mu\text{L}^{-1}$. Solutions were prepared every 24 – 72 hours and stored at 4°C when not in use.

Comment 8: Criteria for excluding animals/sessions

Response 8: See Response to Comment 14.

Comment 9: Statistical power calculations

Response 9: Due to the fact that we could not reasonably estimate effect sizes or variance based on prior literature, as there are no prior examples of intranasal self-administration in animals, we did not perform an *a priori* power calculation.

Comment 10: Method for confirming successful insufflation

Response 10: All sessions were continuously monitored by video captured from a camera that was placed at an angle to the subject, to allow for clear observation of the delivery needle. Successful insufflation was confirmed by direct visual observation during each self-administration session. The experimental apparatus allowed precise alignment of the microfluidic delivery needle with the subject's nostril. Upon delivery, animals reliably inhaled the droplet within 1–2 seconds, and failure to do so was rare. The small volume, delivery location, and immediate behavioral responses (e.g., sniffing or licking) provided consistent behavioral confirmation of successful nasal insufflation. This has been added to the methods section of the manuscript. Moreover, we have now added additional experiments showing blood cocaine concentrations following an intranasal self-administration session, confirming that cocaine was being ingested in a manner proportional to the amount delivered (**Figure 2**).

Comment 11: Initial response curves after the four baseline sessions are complete

Details on how animals were run; 5 simultaneously? Same time of day? Given small n (5), perhaps there is a limitation of the number of animals that can be run concurrently

Response 11: Experiments in the original submission were performed with just two behavioral apparatuses, which were constructed in-house based on earlier pilot experiments / optimization. Thus, animals were run in multiple rounds of two over the course of the day. In the interim, we have scaled up the setup to six, allowing for greater throughput.

Comment 12: Github link in supplement delivers 404 error

Response 12: We apologize for the error, it has been corrected.

Comment 13: Temperature and humidity range in housing and experimental rooms

Response 13: All animals were housed in temperature- and humidity-controlled rooms maintained at $22 \pm 2^\circ\text{C}$ and 40–60% relative humidity, consistent with institutional standards for rodent care. Compliance is verified via continuous temperature and humidity data loggers in the housing room, no values were out of range throughout the course of these studies.

Comment 14:

Data Analysis:

Statistical analyses appear appropriate

Good use of established behavioral economic frameworks

Clear presentation of individual and group data

Some minor concerns:

Sample sizes could be larger for some experiments

Response 14: We have now cross-validated conclusions from the experiments presented in the original submission of the manuscript with additional complimentary experiments, including comparison with saline self-administration controls across multiple fixed ratio requirements (**Figure 3**), analysis of inter-delivery reinforcement times (**Figure S2**), and responding under a progressive ratio schedule of reinforcement (**Figure 4**).

Comment 15: More detail needed on handling of outliers/exclusions

Response 15: The only animals that were excluded from analysis were animals that exhibited minimal responding (i.e., less than 10 lever presses) across the cocaine dose-response sessions, as this indicated lack of engagement with the task. In the initial submission, three animals were excluded based on this criteria, all from the first cohort of animals tested. However, improvements in the apparatus / positioning of the lever have rectified this issue; in the experiments added in the revised manuscript, zero subjects were excluded.

Comment 16: Additional statistical tests for some comparisons would be helpful

Response 16: Additional statistical analyses are now presented in **Figures 2-4**.

Comment 17:

Overall Assessment:

This appears to be a significant methodological advance that could have a broad impact on addiction research. The core findings are well-supported, though some additional details would strengthen the work. The manuscript would benefit from:

More detailed methods

Additional control experiments

Direct comparisons to traditional methods

Expanded discussion of limitations

These revisions would likely be achievable without additional major experiments. The work appears suitable for publication after addressing these points, and would be a notable advance for the field.

Response 17: We thank the reviewer for their time and positive feedback, and helpful suggestions for the manuscript. We feel the responses above have addressed all of the overall concerns listed here, and the manuscript is greatly improved as a result.

Reviewer #4

Comment 1: I co-reviewed this manuscript with one of the reviewers who provided the listed reports. This is part of the Nature Communications initiative to facilitate training in peer review and to provide appropriate recognition for Early Career Researchers who co-review manuscripts.

Response 1: We thank the reviewer for their time and expertise.

References

- Alvites, R. D., Caseiro, A. R., Pedrosa, S. S., Branquinho, M. E., Varejão, A. S. P., & Maurício, A. C. (2018). The nasal cavity of the rat and mouse-source of mesenchymal stem cells for treatment of peripheral nerve injury. *Anatomical Record (Hoboken, N.J.: 2007)*, *301*(10), 1678–1689.
- Barnett, G., Hawks, R., & Resnick, R. (1981). Cocaine pharmacokinetics in humans. *Journal of Ethnopharmacology*, *3*(2–3), 353–366.
- Bravo, R. (2022). Cocaine: An Updated Overview on Chemistry, Detection, Biokinetics, and Pharmacotoxicological Aspects including Abuse Pattern. *Toxins*, *14*.
- Brown, A. R., Branthwaite, H. E., Farahbakhsh, Z. Z., Mukerjee, S., Melugin, P. R., Song, K., Noamany, H., & Siciliano, C. A. (2023). Structured tracking of alcohol reinforcement (STAR) for basic and translational alcohol research. *Molecular Psychiatry*, *28*(4), 1585–1598.
- Bystrowska, B., Adamczyk, P., Moniczewski, A., Zaniewska, M., Fuxe, K., & Filip, M. (2012). LC/MS/MS evaluation of cocaine and its metabolites in different brain areas, peripheral organs and plasma in cocaine self-administering rats. *Pharmacological Reports: PR*, *64*(6), 1337–1349.
- Cone, E. J. (1995). Pharmacokinetics and pharmacodynamics of cocaine. *Journal of Analytical Toxicology*, *19*(6), 459–478.
- Faelens, G., Corriols-Noval, P., & Morales-Angulo, C. (2021). Otolaryngology manifestations of cocaine abuse. *Anales de Otorrinolaringología Mexicana*, *66*, 140–150.
- Isenschmid, D. S., Fischman, M. W., Foltin, R. W., & Caplan, Y. H. (1992). Concentration of cocaine and metabolites in plasma of humans following intravenous administration and smoking of cocaine. *Journal of Analytical Toxicology*, *16*(5), 311–314.
- Javaid, J. I., Fischman, M. W., Schuster, C. R., Dekirmenjian, H., & Davis, J. M. (1978). Cocaine plasma concentration: relation to physiological and subjective effects in humans. *Science (New York, N. Y.)*, *202*(4364), 227–228.
- Jeffcoat, A. R., Perez-Reyes, M., Hill, J. M., Sadler, B. M., & Cook, C. E. (1989). Cocaine disposition in humans after intravenous injection, nasal insufflation (snorting), or smoking. *Drug Metabolism and Disposition: The Biological Fate of Chemicals*, *17*(2), 153–159.
- Lau, C. E., & Sun, L. (2002). The pharmacokinetic determinants of the frequency and pattern of intravenous cocaine self-administration in rats by pharmacokinetic modeling. *Drug Metabolism and Disposition: The Biological Fate of Chemicals*, *30*(3), 254–261.
- Lewis, A. S., Calipari, E. S., & Siciliano, C. A. (2021). Toward Standardized Guidelines for Investigating Neural Circuit Control of Behavior in Animal Research. *ENeuro*, *8*(2), ENEURO.0498-20.2021.
- McGrath, J., McGrath, A., Burdett, J., Shokri, T., & Cohn, J. E. (2020). Systemic pharmacokinetics of topical intranasal cocaine in healthy subjects. *American Journal of Rhinology and Allergy*, *34*(3), 336–341.
- Navarro, M., Ruberte, J., & Carretero, A. (2017). Respiratory apparatus. In *Morphological Mouse Phenotyping* (pp. 147–178). Elsevier.
- Nolan, S. O., Melugin, P. R., Erickson, K. R., Adams, W. R., Farahbakhsh, Z. Z., Mcgonigle, C. E., Kwon, M. H., Costa, V. D., Hackett, T. A., Cuzon Carlson, V. C., Constantinidis, C., Lapish, C. C., Grant, K. A., & Siciliano, C. A. (2025). Recurrent activity propagates through labile ensembles in macaque dorsolateral prefrontal microcircuits. *Current Biology: CB*, *35*(2), 431-443.e4.
- Sanvisens, A., Hernández-Rubio, A., Zuluaga, P., Fuster, D., Papaseit, E., Galan, S., Farré, M., & Muga, R. (2021). Long-term outcomes of patients with cocaine Use Disorder: A 18-years addiction cohort study. *Frontiers in Pharmacology*, *12*, 625610.
- Shannon, M. (1988). Clinical toxicity of cocaine adulterants. *Annals of Emergency Medicine*, *17*(11), 1243–1247.
- Siciliano, C. A., Calipari, E. S., Cuzon Carlson, V. C., Helms, C. M., Lovinger, D. M., Grant, K. A., & Jones, S. R. (2015). Voluntary ethanol intake predicts κ -opioid receptor supersensitivity and regionally distinct dopaminergic adaptations in macaques. *The Journal of Neuroscience: The Official Journal of the Society for Neuroscience*, *35*(15), 5959–5968.
- Siciliano, C. A., Calipari, E. S., Yorgason, J. T., Lovinger, D. M., Mateo, Y., Jimenez, V. A., Helms, C. M., Grant, K. A., & Jones, S. R. (2016). Increased presynaptic regulation of dopamine neurotransmission in the nucleus accumbens core following chronic ethanol self-administration in female macaques. *Psychopharmacology*, *233*(8), 1435–1443.
- Siciliano, C. A., Mauterer, M. I., Fordahl, S. C., & Jones, S. R. (2019). Modulation of striatal dopamine dynamics by cocaine self-administration and amphetamine treatment in female rats. *The European Journal of Neuroscience*, *50*(4), 2740–2749.

- Van Dyke, C., Barash, P. G., Jatlow, P., & Byck, R. (1976). Cocaine: plasma concentrations after intranasal application in man. *Science (New York, N.Y.)*, *191*(4229), 859–861.
- Volkow, N. D., Wang, G. J., Fischman, M. W., Foltin, R., Fowler, J. S., Franceschi, D., Franceschi, M., Logan, J., Gatley, S. J., Wong, C., Ding, Y. S., Hitzemann, R., & Pappas, N. (2000). Effects of route of administration on cocaine induced dopamine transporter blockade in the human brain. *Life Sciences*, *67*(12), 1507–1515.
- Zachry, J. E., Johnson, A. R., & Calipari, E. S. (2019). Sex Differences in Value-Based Decision Making Underlie Substance Use Disorders in Females. *Alcohol and Alcoholism*, *54*(4), 339–341.